
**Probabilistic Fault Displacement Hazard Analysis for North Tabriz Fault**
Mohamadreza Hosseyni[1], Habib Rahimi, [2]
*1. M.Sc. Graduated, Department of Earth Physics, Institute of Geophysics, University of Tehran, Tehran, Iran*
*2. Associate Professor, Department of Earth Physics, Institute of Geophysics, University of Tehran, Tehran, Iran*
*Corresponding author: Habib Rahimi; email: rahimih@ut.ac.ir*
**Abstract:**
The probabilistic fault displacement hazard analysis is one of the new methods in estimating the amount of possible
displacement in the area at the hazard of causal fault rupture. In this study, using the probabilistic approach and
earthquake method introduced by Youngs et al., 2003, the surface displacement of the North Tabriz fault has been
investigated, and the possible displacement in different scenarios has been estimated. By considering the strike-slip
mechanism of the North Tabriz fault and using the earthquake method, the probability of displacement due to surface
ruptures caused by 1721 and 1780 North Tabriz fault earthquakes has been explored. These events were associated
with 50 and 60 km of surface rupture, respectively. The 50-60 km long section of the North Tabriz fault was selected
as the source of possible surface rupture.
We considered two scenarios according to possible displacements, return periods, and magnitudes which are reported
in paleoseismic studies of the North Tabriz fault. As the first scenario, possible displacement, return period, and
magnitude was selected between zero to 4.5; 645 years and Mw~7.7, respectively. In the second scenario, possible
displacement, return period and magnitude were selected between zero to 7.1, 300 years, and Mw~7.3, respectively.
For both mentioned scenarios, the probabilistic displacements for the rate of exceedance 5% in 50, 475, and 2475
years for the principle possible displacements (on fault) of the North Tabriz fault have been estimated. For the first
and second scenarios, the maximum probabilistic displacement of the North Tabriz fault at a rate of 5% in 50 years is
estimated to be 186 and 230 cm. Also, mentioned displacements for 5% exceedance in 475 years and 2475 years in
both return periods of 645 and 300 years, are estimated at 469 and 655cm.
**Keywords**: Surface rupture, Hazard, probabilistic fault displacement, North Tabriz fault, Iran.

**1- Introduction**
Earthquakes, not only because of earth-shaking but also because of surface ruptures, are a serious threat to many
human activities. Reducing earthquake losses and damages requires predicting the amplitude and location of ground
movements and possible surface displacements in the future. Fault displacement hazard assessments are based on
empirical relationships obtained using historical seismic rupture data. These relationships evaluate the probability of
co-seismic surface slip of ruptures on fault (primary) and outside the fault (distributed) for different magnitudes and



distances to the causal fault. In addition, these relationships make it possible to predict the extent of fault slip on or
near the active fault (Stephanie Baiz et al., 2019).

36          A way to reduce the effects of fault rupture hazards on a structure is to develop the probability of fault

displacement. This approach can be taken into account the rate of exceedance of different displacement levels of the
event under a structure, along with a displacement hazard curve (Youngs et al., 2003). In surface displacement hazard
studies, non-tectonic displacements such as fault creep, aftershock, soil liquefaction, and landslide are not considered
(Petersen et al., 2011). So far, fault displacement data have been collected and analyzed by several researchers to
evaluate the fault rupture properties. Investigation of fault displacement and extraction of experimental relationships
between rupture length and magnitude, rupture length and fault mechanism, maximum fault displacement, average
fault displacement, and other cases are investigated by Wells and Coppersmith (1993 and 1994) and reviewed by
Petersen and Wesnousky (1994). To be considered, each earthquake causes a superficial shaking at the site, but each
earthquake does not cause a surface rupture in the area. Therefore, only the data of earthquakes that have caused the
rupture in the region are used to obtain the attenuation relationships (Youngs et al., 2003). Principal displacements are
considered primary ruptures that occur on or within a few meters of the active fault. Distributed displacements outside
the fault are causative and usually appear as discontinuous ruptures or shears distance several meters to several
hundred kilometers from the fault trace. The principal and distributed displacements are introduced as net
displacements derived from horizontal and vertical displacements (Petersen et al., 2011).
A method for estimating the probabilistic fault displacement hazard for strike-slip faults in the world has been
presented, mapped due to the impact of fault displacement hazard to the fault trace type and the complexity of this
effect and hazard of fault displacement for strike-slip faults studied (Petersen et al., 2011). The North Tabriz fault has
a high level of danger by passing through the 5th district of Tabriz city, and in case of possible surface rupture, it will
lead to many damages in this residential area. With 150,000 people and 32,000 square kilometers, this region has
essential areas such as Baghmisheh, Elahieh, Rashidieh town, etc. (Figure 1).
In this study, based on the results of a paleoseismic study reported by Hesami et al. (2003) on the North Tabriz fault,
the section with a length of 50 - 60 km was considered as a source of possible rupture in the future. To describe the
possible behavior of the displacement rupture hazard of the North Tabriz fault, sites at distances of 50 m from each
other and cells with dimensions of $25 \times 25$ m$^2$ on fault trace were considered, which is shown in Figures 1. Also,
according to the study of Petersen et al. (2011), the trace of the North Tabriz fault was considered as a simple trace
due to the absence of large instrumental earthquakes that are associated with surface rupture. Many studies have been
done on the historical displacements of the North Tabriz fault. According to the results of paleoseismic studies reported
by Hesami et al. (2003) and Ghasemi et al. (2015), the probabilistic displacement is between zero to 4.5 and zero to
7.1 m, respectively. The magnitude and return period of large earthquakes are considered 645 years with Mw ~7.7
and 300 years with Mw~7.3 according to Mousavi et al. 2014 and Dejamour et al., 2011, respectively.
In this study and the first step, probabilistic fault displacement and annual rate of exceedance of displacement for two
given scenarios (645 years with Mw ~7.7) and (300 years with Mw ~7.3) have been achieved by considering 5%
exceedance rate in 50, 475, and 2475 years at the site with geographical coordinates (38.096, 46.349). In the second



step, due to the passage of the North Tabriz Fault through the city of Tabriz, considering a 2 km long section from the
North Tabriz Fault, the probabilistic displacement has been estimated, and the probabilistic displacement 2D map is
explored.


**2-  Seismotectonic**

76        With over two million people and an area of 167 square kilometers in northwestern Iran, Tabriz is one of the

most populated cities in the country that has experienced devastating earthquakes throughout history. One of the main
problems of Tabriz City is the proximity of the city to the North Tabriz fault and the expansion of constructions around
it. Based on the reported historical earthquakes by Berberian and Arshadi (1979), since 858 AD., this city and the
surrounding area have experienced several large and medium destructive earthquakes.
The focal mechanism of earthquakes in northwestern Iran and southeastern Turkey shows that the convergence
between the Saudi and Eurasian plates becomes depreciable during right-lateral strike-slip faults. The strike-slip fault
is the southeastern continuation of the North Anatolian Fault into Iran, consisting of discontinuous fault sections with
a northwest-southeast extension (Jackson and Mackenzie et al., 1992). Some of these fault fragments have been
ruptured and left deformed along with the earthquakes in 1930, 1966, and 1976 (Hesami et al., 2003).
Nevertheless, the North Tabriz fault is one of the components of this right-lateral strike-slip system, which has not
had a major earthquake during the last two centuries. Among the many historical earthquakes in the Tabriz region,
only three devastating earthquakes with a magnitude of Ms~7.3 in 1042, 1721, and 1780 with a magnitude of Ms~7.4
had been associated with a surface rupture along the North Tabriz fault (Hesami et al., 2003). The 1721 and 1780 AD
earthquakes were along with at least 50 and 60 km of surface rupture (about 40 km overlap), respectively. Berberian
et al., 1997 believe that large earthquakes along the North Tabriz fault are concentrated at specific times and spatially
related.
The occurrence of the 1976 Chaldoran earthquake in Turkey, which was accompanied by about 55 km of fractures,
indicates that the length of the surface fracture caused by historical earthquakes in this region probably varies from
about 50 to 60 km (Toxos et al., 1977). A more detailed study of the temporal distribution of earthquakes in Tabriz by
Berberian and Yates (1999) also shows the cluster distribution of earthquakes over time. Due to the absence of seismic
events for more than 200 years in the Tabriz area (decluttering period), the study area has passed the final stages of
stress storage, and it is ready to release the stored energy. Therefore, Hesami et al., 2003 investigated the Spatial-
temporal concentration of earthquakes associated with the North Tabriz fault. Based on paleontological seismic studies
on the western part of the North Tabriz fault, Hesami et al., 2003 introduced four earthquakes that occurred
continuously on the western part of the North Tabriz fault. The return periods of these earthquakes were suggested to
be 821 ± 176 years. The amount of right-lateral strike-slip displacement, during each seismic event, of the North
Tabriz fault, has been estimated at 3.5 to 4.5 m. In addition, Berberian et al., 1997 considered the possibility of




fracturing all parts of the North Tabriz fault at once and mentioned it as one of the critical issues in the earthquake
hazard for the Tabriz city and the northwestern region of Iran.

**3- Methodology of probabilistic fault displacement hazard analysis**
Probabilistic seismic hazard analysis has been used since its development in the late 1960s and early 1970s
to assess shaking hazards and to establish seismic design parameters (Cornell, 1968 and 1971). A method for analyzing
the hazard of probabilistic fault displacement was introduced in two approaches of earthquake and displacement
(Youngs et al., 2003). This method was first proposed to estimate the displacement of Yucca Mountain faults, which
were the landfill of nuclear waste (Stepp et al., 2001). Then, the probabilistic fault displacement hazard analysis
method was introduced for an environment with normal faults, and the probability distributions obtained for each type
of fault in the world can be used in areas with similar tectonics (Youngs et al., 2003).
The earthquake approach is similar to the analysis of probabilistic seismic hazard related to displacement, features
such as faults, partial shear, fracture, or unbroken ground at or near the ground surface so that the attenuation
relationships of the fault displacement replace the ground shaking relationships. In the displacement approach, without
examining the rupture mechanism, the displacement characteristics of the fault observed at the site are used to
determine the hazard in that area.
The occurrence rate of displacements and the distribution of fault displacements are obtained directly from the fault
characteristics of geological features (Youngs et al., 2003). To calculate the rate of exceedance in the earthquake
approach, similar to probabilistic seismic hazard analysis relationships were used. The rate of exceedance, $v_k(z)$, is
calculated according to the Cornell relationship (1968 and 1971) as follows (Youngs et al., 2003):

$$v_k(z) = \sum_n \alpha_n(m^0) \int_{m^0}^{m_n^u} f_n(m) [\int_0^\infty f_{kn}(r|m). P^*(Z>z|m,r).dr].dm \qquad (1)$$

In which the ground motion parameter, (Z), (maximum ground acceleration, maximum response spectral acceleration)
exceed the specified level (z) at the site (k). Considering Equation (1) and to calculate the exceedance rate of
displacement (D) from a specific value (d), the displacement parameter replaces the parameters of ground motion
(Youngs et al., 2003):

$$v_k(d) = \sum_n \alpha_n(m^0) \int_{m^0}^{m_n^u} f_n(m) [\int_0^\infty f_{kn}(r|m). P^*(D>d|m,r).dr].dm \qquad (2)$$

The expression P (D>d|m,r) is the "attenuation function" of the fault displacement at or near the earth's surface. This
displacement attenuation function is different from the usual ground motion attenuation function and includes the
multiplication of the following two probabilities (Youngs et al., 2003):

$$P_{kn}^*(D>d|m, r) = P_{kn}(Slip|m, r). P_{kn}(D>d| m, r, slip) \qquad (3)$$

Which *D* and *d* are the Displacement on fault (principal fault) and displacement on the outside of the fault (distributed
fault), respectively. (x, y) are considered as coordinates of the site. r, $z^2$, I, L, and s are the vertical distance from the



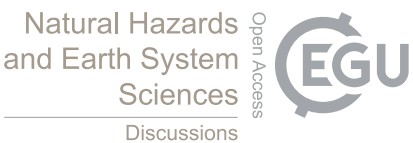
fault,  area, the distance of site on fault rupture to the nearest rupture, the total length of the fault surface rupture, and
the rupture distance to the end of the fault, respectively. The definition of these variables is shown in figure (2).
The following Equation has been used to obtain the exceedance rate of probabilistic displacement due to the principal
fault (on fault) (Petersen et al., 2011):

$$\lambda(D \geq D_0)xyz = \qquad\qquad\qquad\qquad (4)$$

$$\alpha(m) \int_{m,s} f_{M,S}(m,s) P[sr \neq 0|m] * \int_{r} P[D \neq 0|z, sr \neq 0] * P\left[D \geq D_0 \Big|{}^l/_L, m, D \neq 0\right] f_R(r) dr dm ds$$

The magnitude of the earthquake is indicated by m. In relation 4 and to assess the displacement hazard due to fault
rupture, the probability density functions that describe displacement potential due to earthquakes on or near a rupture,
as well as the probabilities that the potential for non-zero ruptures are used (Petersen et al., 2011). In the following,
each of the parameters for estimation of probabilistic fault displacement hazard is described.
**3-1 Probability density function**
The probability density function $f_{M,s}(m,s)$ determines the magnitude of the earthquake and the location of
the ruptures on a fault. Since the magnitude and the rupture position on the causal fault are correlated, a probabilistic
distribution is used to calculate these parameters. In the next step, the variability in the rupture location is considered.
A probability density function $f_R(r)$ is considered to define the area of perpendicular distances (r) to the site to different
potential ruptures (Petersen et al., 2011).
**3-2 Probabilities**
Probability P [SR ≠ 0 | M] is the ratio of cells with rupture on the principal fault to the total number of cells
considered. Therefore, the probability of surface rupture P [SR ≠ 0 | M] is considered due to a certain magnitude M
due to faulting. According to studies of Wells and Coppersmith (1993), due to the formulation of empirical
relationships between different fault parameters, probability has been obtained for different faults in the world, such
as strike-slip, normal, and revers. Therefore, in hazard analysis of fault displacement, it is necessary to investigate the
possibility of surface rupture with magnitude (M) on the ground so as a result, the equation (5) introduced by Wells
and Coppersmith (1993) can be used. According to this relation, the coefficients a and b are constant, and strike-slip
faults with -12.51 and 2.553 have been reported. This relationship has a 10% probability for the size of Mw~5 and a
95% probability of surface rupture for a magnitude of Mw~7.5 ((Rizzo et al., 2011).

$$P[sr \neq 0|m] = \frac{e^{a+bm}}{1+e^{a+bm}} \qquad\qquad\qquad (5)$$

This rupture probability was used to estimate the exceedance rate of displacement because of earthquakes such as
Loma-Prieta in 1989 with a magnitude of Mw~6.9 and Alaska in 2002 with a magnitude of Mw~6.7. These
earthquakes did not cause rupture to reach the earth's surface. Therefore, these two earthquakes did not cause surface



deformation and are considered non-tectonic phenomena (Petersen et al., 2011). The expression $P[D \neq 0|z, sr \neq 0]$
indicates the probability of non-zero displacement at a distance r from the rupture in an area of size $z^2$ and due to the
magnitude event m associated with the surface rupture. The probability $P[D \geq D_0|{}^{l}/_{L}, m, D \neq 0]$ for displacements
more significant than or equal to the value given at this site is intended for the principal displacement (on fault). This
probability is obtained by integrating around a log-normal distribution (Petersen et al., 2011).

**3-3 Rate parameter α(m):**
When the potential magnitude of an earthquake with a certain magnitude is modeled, it is possible to estimate how
often these ruptures occur. The, α (m), rate parameter used describes the frequency of repetition of these earthquakes
in this model. This parameter is a function of magnitude and can only function as a single rupture function or a function
of cumulative earthquakes above the magnitude of the minimum importance in engineering projects (Youngs et al.,
2003). This parameter is usually based on slip rate, paleoseismic rate of large earthquakes, or historical fault rate
earthquakes and is described in earthquake units per year. By removing the α (m) parameter from Equation (4), the
Deterministic Fault Displacement Hazard can be estimated (Petersen et al., 2011).

**3-4 Cell size:**
In calculating the hazard of principal fault displacements, as shown in Eq. (4), by changing the size of the cells, the
level of hazard will not change and this parameter can be examined by the availability of principal displacement data
in the study area. In calculating the hazard of distributed rupture (distributed displacement), considering the method
of Youngs et al. (2003), by modeling secondary displacements up to a distance of 12 km from the fault, the probability
of surface rupture was investigated. According to studies by Petersen (2011), the relationship between the calculations
of the probability of rupture of the principal faults (5), in calculating the probability of rupture of the distributed faults
became the following relationship (Petersen et al., 2011):

Ln (p)= a(z) ln(r) + b(z)                                                          (6)

The values of the coefficients used for the cell sizes of 25×25 to 200×200 m$^2$ in the above relationship are given in
Table 1 (Petersen et al., 2011).

**3-5 Surveying accuracy**

190       The accuracy of fault location is a function of geological and geomorphic conditions that play an essential

role in diagnosing and interpreting a geologist in converting this spatial information into geological maps and fault
geographic information systems. A fault map is generated using aerial photography imagery, interpretation of fault
patterns from geomorphology, and conversion of fault locations into a base map. In many cases, identifying the
location and trace of the fault may be difficult because sediments and erosion may obscure or cover the fault surface,
leading to more uncertainty in identifying the actual location of the fault. Therefore, trace mapped faults are divided
into four categories: accurate, approximate, inferred, and concealed, based on how clearly and precisely they are
located (Petersen et al., 2011).



A practical example shows that an active fault with large earthquakes repeated over several hundred years, fault
rupture hazard analysis should be one of the critical topics considered for the design of structures or pipelines that are
close to this fault, and if this the fault has a complex or straightforward trace, avoiding the fault from the constructor
to a distance of 150 and 300 meters, respectively. Table 2 summarizes the standard deviations for the displacements
observed in strike-slip earthquakes for different classifications of mapping accuracy (Petersen et al., 2011). According
to the exponential values obtained from these fitting equations, the mean displacement will be obtained. The following
Equation has been used to obtain the mean displacement (Petersen et al., 2011):

$$D_{mean} = e^{\mu + \sigma^2/2} \qquad\qquad (7)$$

**3-6 Epistemic and Aleatory uncertainty**
There are uncertainties about the quality of mapping and the complexity of the fault trace that lead to epistemic
uncertainty at the site of future faults. The probability density function for r includes both epistemic and aleatory
components. Displacements on and off the principal fault can include components of epistemic uncertainty and
random variability. Epistemic uncertainty is related to displacement measurement errors along fault rupture. Random
variability is related to the natural variability in fault displacements between earthquakes. However, the measured
variability in ruptures involves epistemic mapping uncertainties because there is currently no data to separate these
uncertainties. In addition, epistemic uncertainty of location is introduced due to limitations in the accuracy of basic
maps or images and the accuracy of the equipment used to transfer this information to the map or database (Petersen
et al., 2011).

**3-7 Attenuation relationship of strike-slip faults**

In this study, to estimate the probabilistic displacement of the North Tabriz fault, the attenuation relationship of
Petersen et al. (2011) has been used. The rupture displacement data obtained from the principal fault are scattered but
are generally the most scattered near the fault rupture center and decrease rapidly at the end of the rupture. In some
earthquakes, including the Borgo Mountain earthquake in 1968, the most significant displacement was observed near
the end of the fault surface rupture (Petersen et al., 2011). Many of the collected surface rupture data behave
asymmetrically ruptured (Wesnousky et al., 2008). However, there is currently no way to determine surface rupture
areas that have larger displacements. Thus, the distribution of asymmetric displacements along the length of a fault
will define more considerable uncertainties, especially near the end of the fault rupture (Petersen et al., 2011). To
determine the displacement distribution, the principal fault, two different approaches were introduced by Petersen et
al. (2011). In the first approach, the best-fit equations using the least-squares method related to the natural logarithm
of the displacement ratio of magnitude and distance were developed in a multivariate analysis (Paul Rizzo et al., 2013).
In the second approach, the displacement data is normalized by the average displacement as a distance function. In
normalized analysis, magnitude is not directly considered but influences calculations through the presence of
magnitude in the mean displacement, which is calculated through the studies of Wells and Coppersmith (1994). Three


models (bilinear, elliptical and quadratic) were considered to provide the principal fault displacement in multivariate
and normalized analysis (Petersen et al., 2011). However, in multivariate analysis, the three introduced models have
the same aleatory uncertainty, and there is no clear basis for preferring one model to other models. As a result, in the
probabilistic displacement hazard analysis, all three models with the same weights were used according to Table 3.
The results obtained from the multivariate analysis were preferred to the normalized analysis because, in the
normalized analysis, the stochastic uncertainty of calculating the mean displacement from the Wells and Coppersmith
(1994) study is added to the stochastic uncertainty of the results of the Petersen attenuation relationships (Paul Rizzo
et al., 2013). In this study, multivariate analysis and probabilistic displacement estimation have been used in the three
mentioned models. The Equation of the three models is obtained in the multivariate method as shown in Table 3, and
5% uncertainty was considered in the modeling of the strike-slip displacement data (Petersen et al., 2011):

**4 Results and Discussions**
**4-1 Probabilistic displacement and rate of exceedance**
In the first step, assuming the possible surface rupture of the North Tabriz fault (50 to 60 km), displacement and the
annual exceedance rate are estimated by considering one of the sites located on the Tabriz fault trace related to the
total segment as shown in Figure 1. Considering the return periods of 645 and 300 years, the probabilistic
displacements of the North Tabriz fault are assumed 4.5 and 7.1 m according to Hesami et al. (2003) and Ghasemi et
al. (2015), respectively.
Given the 4.5 m probabilistic displacements reported by Hesami et al. (2003), maximum displacement for the return
period of 645 years at an exceedance rate of 5% for 50, 475, and 2475 years are estimated 186, 469, and 469cm,
respectively. The maximum displacement for a return period of 300 years is calculated at 230, 469, and 469cm. These
amounts of displacement were observed for the return period of 645 and 300 years at a distance of 60-100 and 60-80
meters from the selected site respectively, as shown in figure (3a).
Also, by considering the 7.1 m probabilistic displacements reported by Ghasemi et al. (2015), maximum displacement
for the return period of 645 years with an exceedance rate of 5% in 50, 475, and 2475 years are estimated 186, 655,
and 655cm, respectively. The maximum displacement for a return period of 300 years is calculated at 230, 655, and
655 cm.  These amounts of displacement were observed for the return period of 645 and 300 years at a distance of 60-
80 and 40-80 meters from the selected site respectively, as shown in figure (3b).

**4-2 Comparison of different fitting models**
As mentioned, the fitting models (bilinear, elliptical, and quadratic) have similar uncertainties, and this section
compares the displacements obtained from these models. In this study, the bilinear model is used to obtain probabilistic
displacements. The values of the probabilistic displacements obtained for the models (bilinear, elliptical, and
quadratic) are shown in Figure 4.





**4-3 Annual exceedance rate of 5% in 50 years**
Assuming the trace of the North Tabriz fault and considering the bilinear model and the return period of 645 and 300
years, the annual rate of exceedance for the two displacement scenarios of 4.5 and 7.1m has been examined.
In this comparison, an annual rate of exceedance of 5% in 50 years for both displacement scenarios of 4.5 and 7.1m,
at distances 64 and 120m from the assumed site, has been examined in figure (5). In the case of D=4.5m, the annual
rate of exceedance of D=4m at distances of 64 and 120 m for 645 years is $1.81 \times 10^{-4}$ and $7.51 \times 10^{-6}$ and for 300 years
$2.17 \times 10^{-4}$ and $1.32 \times 10^{-5}$ as shown in (6a). In the case of D=7.1m, the annual rate of exceedance of D= 4m at distances
of 64 and 120m for 645 years is $1.81 \times 10^{-4}$ and $1.81 \times 10^{-4}$ and for 300 years $1.81 \times 10^{-4}$ and $1.81 \times 10^{-4}$ as shown in (6b).

**4-4 Probabilistic displacement of North Tabriz fault**
By examining the trace of the North Tabriz fault, due to the passing from the fifth region of Tabriz city, estimating
the probabilistic displacement in the region is essential and predicting the areas with a higher level of hazard is an
important matter. Considering a 2 km long section of the North Tabriz fault according to Figures 6, 7, and 8, the two-
dimensional probabilistic displacements for the North Tabriz fault have been estimated. To estimate the probabilistic
displacement, two scenarios (Mw7.7, 645years) and (Mw7.3, 300years) were considered. Figure 6 shows the
probabilistic displacement of the two scenarios mentioned for the 5% exceedance rate in 50 years, by the blue color
spectrum. The probabilistic displacements for the 4.5 and 7.1 m displacements for the first scenario are shown in
Figures 6a and 6b, respectively, and for the second scenario, in Figures 6c and 6d, respectively. The probabilistic
displacement values for the second scenario have a higher level of hazard that can be seen at greater distances from
the assumed sites. The probabilistic displacement of the two scenarios for the 5% increase rate at 475 and 2475 years
in Figures 7 and 8, respectively, using the blue to red color spectrum is shown.
Due to the lack of high magnitude instrumental earthquakes and surface ruptures, the discussion about the probabilistic
failure level of this active fault is uncertain in the future. As a result, one of the ways to reduce the level of damage
and financial and human losses is to avoid construction around this fault trace due to several terrible historical
earthquakes.

**5 Conclusion**
Assuming the mechanism of strike-slip and trace of Tabriz fault as a simple trace, and considering two scenarios
(Mw~7.7, 645yrs), and (Mw~7.3, 300yrs) and a fault section with a length of 50 - 60 km, the probabilistic
displacement of the North Tabriz fault was estimated. Furthermore, considering the reported approach by Petersen
(2011), the probabilistic displacements for an exceedance rate of 5% in 50, 475, and 2475 years for the principal
probabilistic displacements (on fault) of the North Tabriz fault have been explored. The obtained results in this study
can be summarized as follows.


1- We considered two scenarios according to possible displacements, return periods, and magnitudes which are reported in paleoseismic studies of the North Tabriz fault.

2- In the first scenario, possible displacement, return period and magnitude were selected between zero to 4.5; 645 years and Mw~7.7, respectively. In the second scenario, possible displacement, return period and magnitude were selected between zero to 7.1, 300 years, and Mw~7.3, respectively.

3- For both above- mentioned scenarios, the probabilistic displacements for the rate of exceedance 5% in 50, 475, and 2475 years for the principle possible displacements (on fault) of the North Tabriz fault have been estimated. For the first and second scenarios, the maximum probabilistic displacement of the North Tabriz fault at a rate of 5% in 50 years is estimated to be 186 and 230 cm.

4- Maximum displacements for 5% exceedance in 475 years and 2475 years in both return periods of 645 and 300 years are estimated at 469 and 655cm.

5- In this study, the probability displacement values of the North Tabriz fault have been obtained without considering the dip, depth, and rake of the fault, which has caused the same displacement values in the north and south plane of the fault. In future studies, it is possible to investigate the geometric properties of the source producing surface rupture and reduce the recognition uncertainty in the method of probabilistic fault displacement hazard analysis.

6- The lack of large instrumental earthquakes in northwestern Iran leads to more significant epistemic uncertainty in the obtained values. Due to the passing of the North Tabriz fault through the residential area of Tabriz and destructive historical earthquakes, it is crucial to estimate the possible future displacements of this fault.

**Conflicts of interests**

The authors declare that they have no known competing financial interests or personal relationships that could have appeared to influence the work reported in this paper.

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

**418      List of figures:**

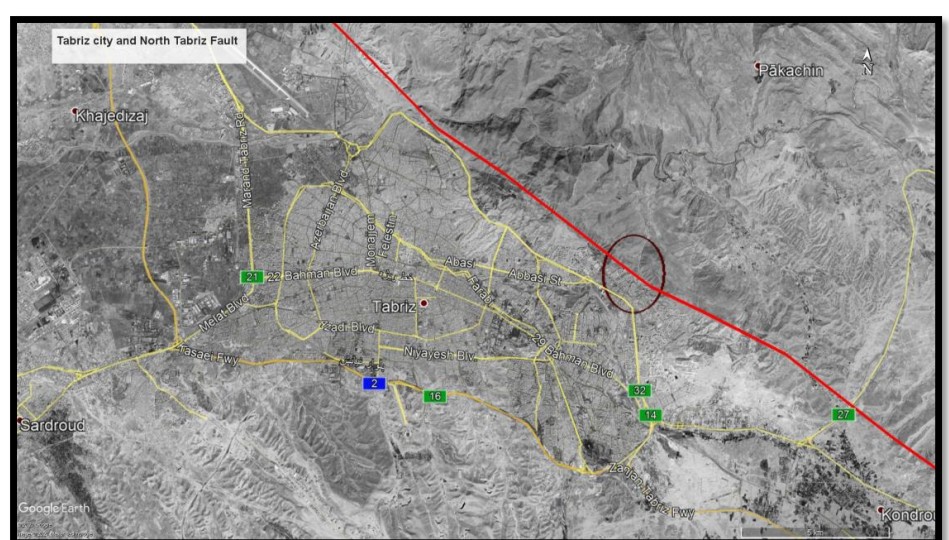


420                                               (a)

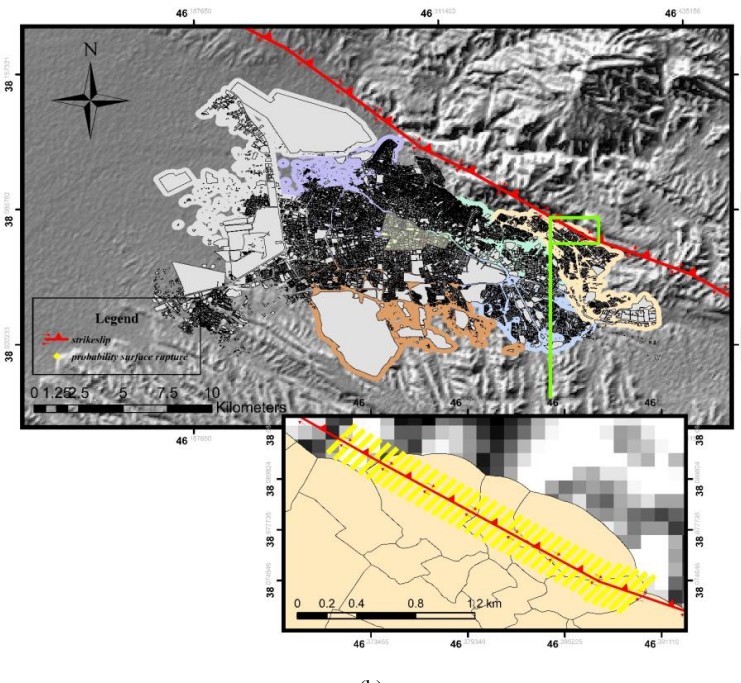


422                                               (b)

Figure 1. North Tabriz Fault and Tabriz city (a), Part of the North Tabriz fault considered in this study and perpendicular profiles
(b). Figure a and b are generated using Google Earth with Digital Globe imagery (© Google Earth 2021).

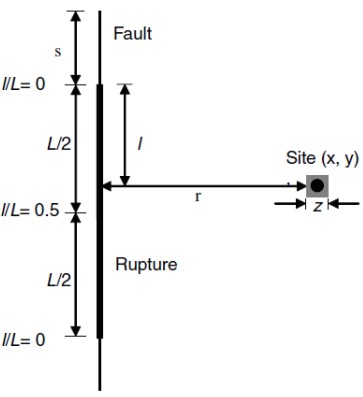


Figure 2. Definition of the variables used in fault rupture analysis: x and y Site coordinates, z Dimensions of the area intended to calculate the probability of fault rupture at the site (for example, dimensions of the building foundation), r: the distance from the site to the fault trace, ratio l/L: the distance from the fault so that l is the measured distance from the nearest point on the rupture to the nearest end of the rupture, L: the total length of the rupture and s: the distance from the end of the rupture to the end of the fault (Petersen et al., 2011).



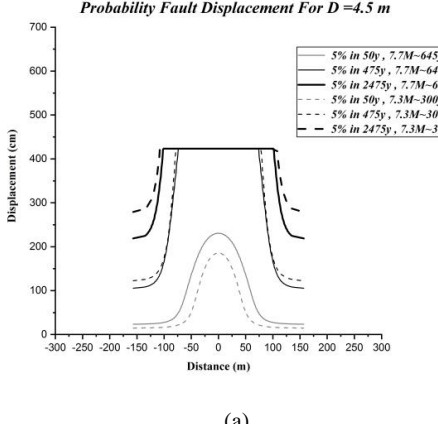

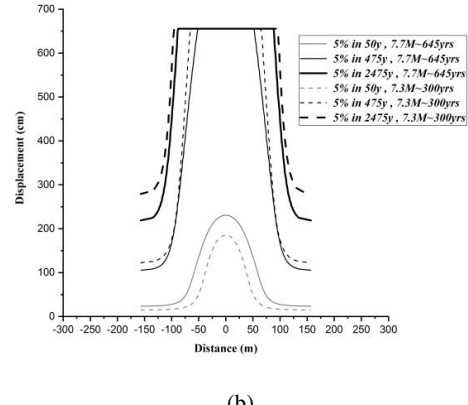

(a)

(b)


Figure 3. Comparison of probability displacement, 5% exceedance rate in 50, 475, and 2475 years for a) D=4.5 m b) D=7.1 m






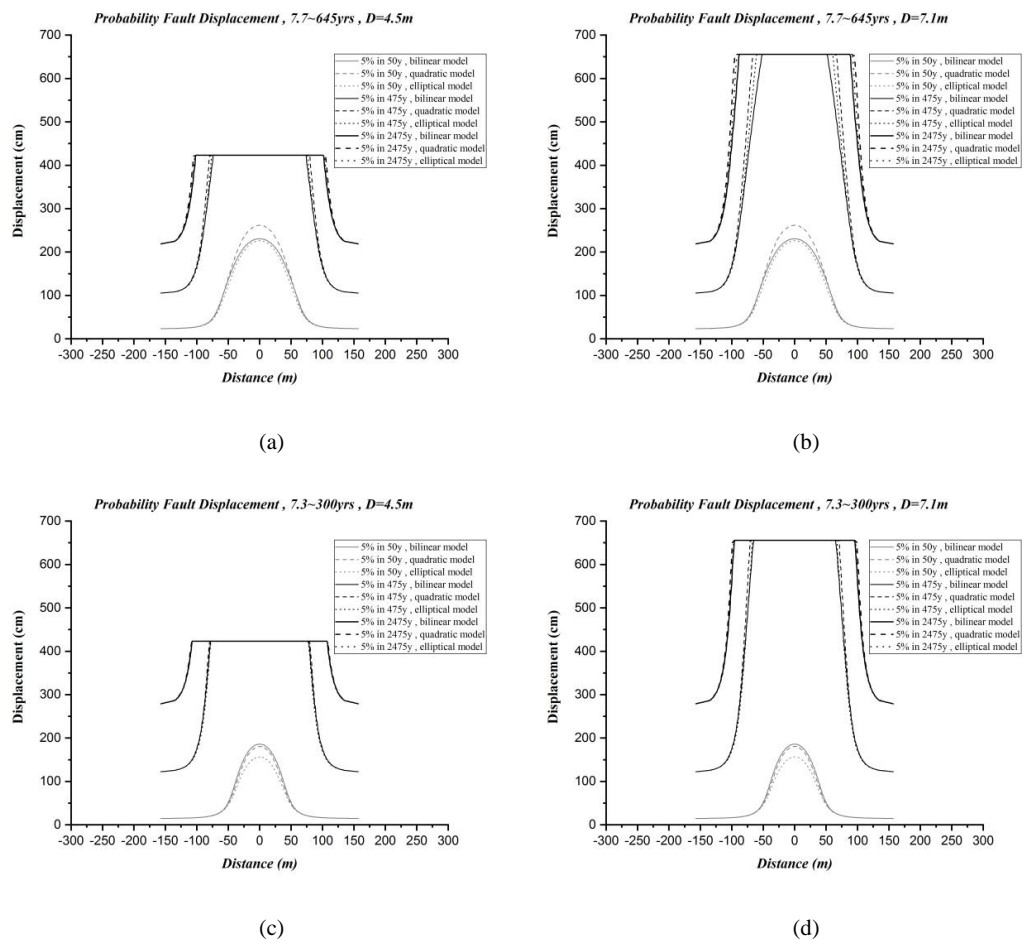

(a)

(b)

(c)

(d)

Figure 4. Comparison of probability displacement, different fitting models for a) 645-year return period and D=4.5 m, b) 645-year return period and D= 7.1m, c) 300-year return period and 4.5 m, d) return period 300- years, and D=7.1 m




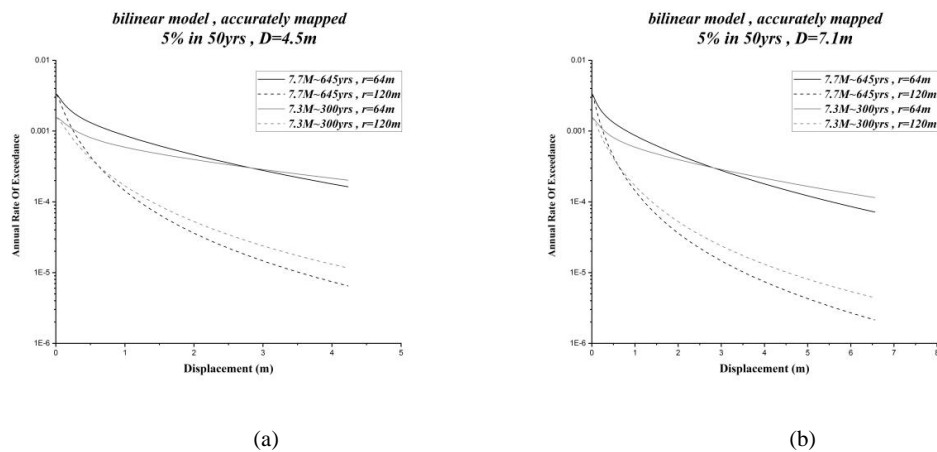

(a)                                                              (b)

446          Figure 5. Comparison of the annual rate of exceedance of displacement for a) D=4.5 m displacement, b) D=7.1 m displacement



















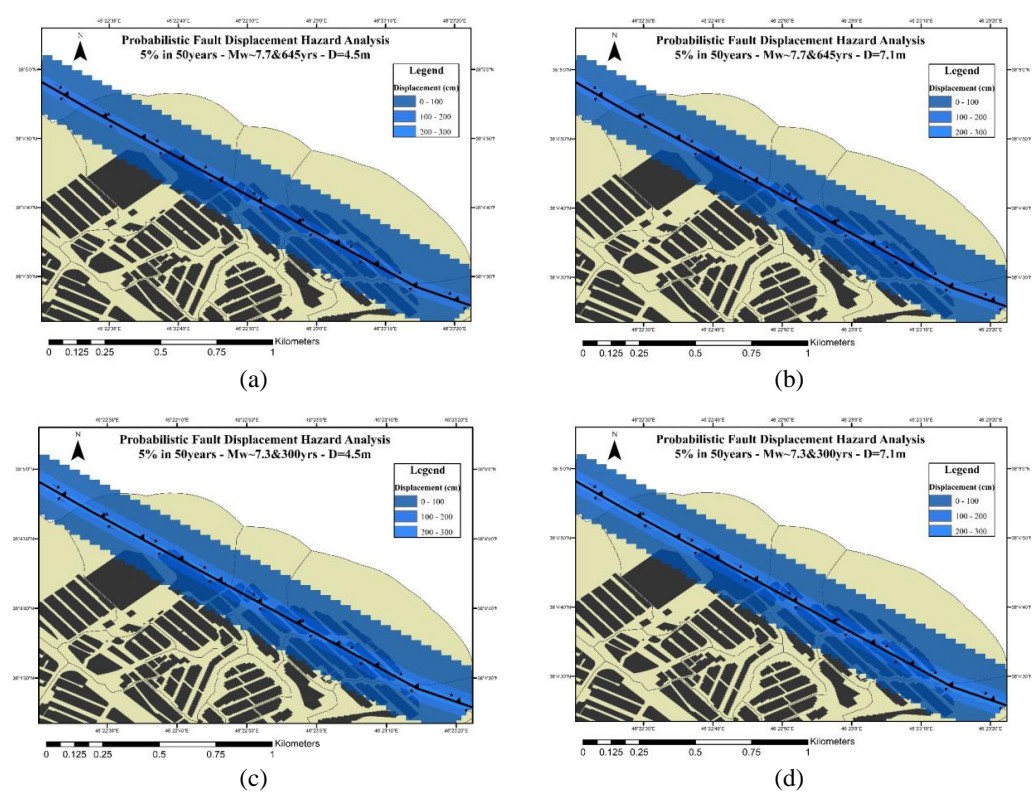

Figure 6. Probability Displacement of 5% in 50, a)Mw~7.7 and return period of 645yrs for D=4.5m, b) Mw~7.7 and return period of 645yrs for D=7.1m, c) Mw~7.3 and return period of 300yrs for D=4.5m and d) Mw~7.3 and return period of 300yrs for D=7.1m

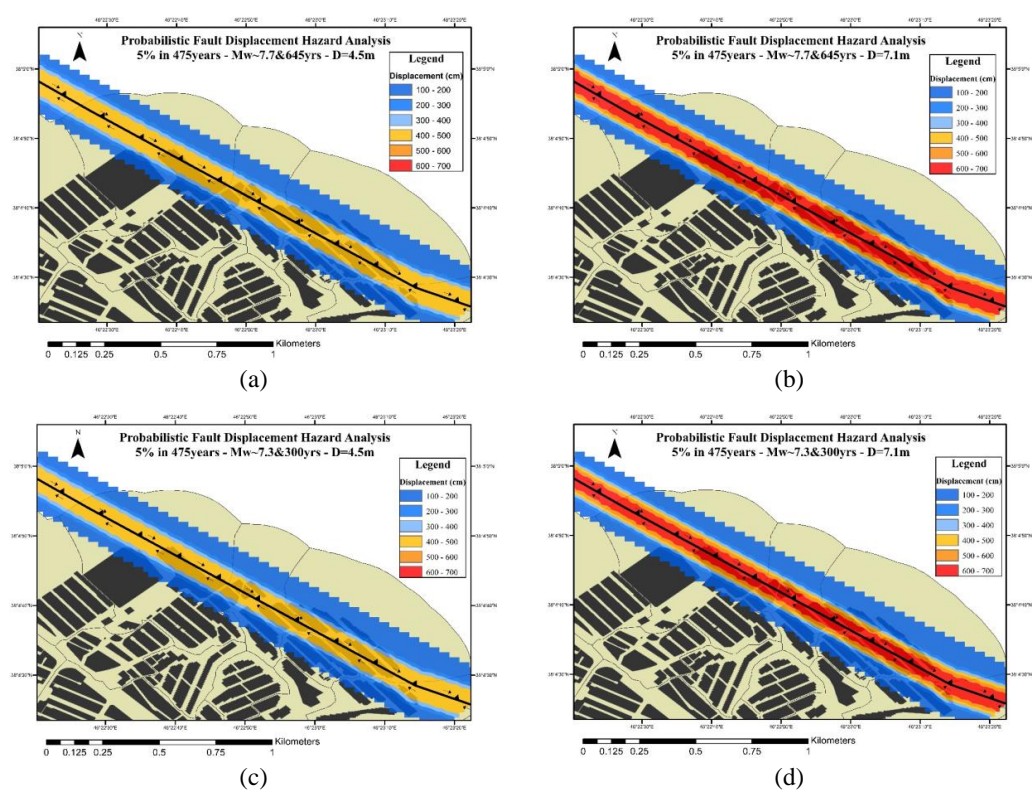

(a)

(b)

(c)

(d)

Figure 7. Probability Displacement of 5% in 475, a)Mw~7.7 and return period of 645yrs for D=4.5m, b) Mw~7.7 and return period of 645yrs for D=7.1m, c) Mw~7.3 and return period of 300yrs for D=4.5m and d) Mw~7.3 and return period of 300yrs for D=7.1m

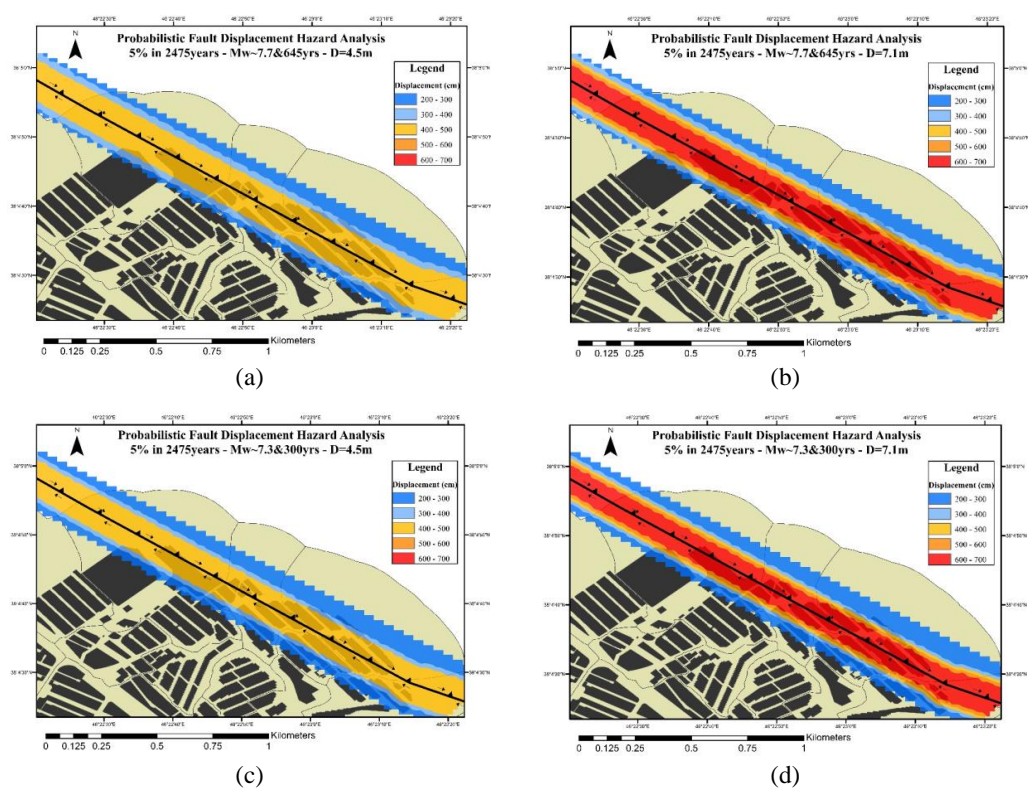

(a)  (b)

(c)  (d)


Figure 8. Probability Displacement of 5% in 2475, a)Mw~7.7 and return period of 645yrs for D=4.5m, b) Mw~7.7 and return period of 645yrs
for D=7.1m, c) Mw~7.3 and return period of 300yrs for D=4.5m and d) Mw~7.3 and return period of 300yrs for D=7.1m













List of Tables:

Table 1. Probability of distributed rupture for different cell sizes (Petersen et al., 2011)

| No. | Cell Size (m$^2$) | a(z) | b(z) | Standard Deviation(ϭ) |
|---|---|---|---|---|
| 1 | 25×25 | -1.1470 | 2.1046 | 1.2508 |
| 2 | 50×50 | -0.9000 | 0.9866 | 1.1470 |
| 3 | 100×100 | -1.0114 | 2.5572 | 1.0917 |
| 4 | 150×150 | -1.0934 | 3.5526 | 1.0188 |
| 5 | 200×200 | -1.1538 | 4.2342 | 1.0177 |







Table 2. Summary of mapping accuracy: The measured distance from the mapped fault trace to the observed surface rupture (Petersen et al., 2011)

| Mapping Accuracy | Mean (m) | One-Sided Standard Deviation (m) | Two-Sided Standard Deviation on Fault (m) |
|---|---|---|---|
| ALL | 30.64 | 43.14 | 52.92 |
| Accurate | 18.47 | 19.54 | 26.89 |
| Approximate | 25.15 | 35.89 | 43.82 |
| Concealed | 39.35 | 52.39 | 65.52 |
| Inferred | 45.12 | 56.99 | 72.69 |

Table 3. Different Models Used in Principal Fault Attenuation Relationships (Petersen et al., 2011)

| Analysis Type | Model | Weight |
|---|---|---|
| Multivariate | **BILINEAR**<br>$\ln(D)=1.7969Mw+8.5206(l/L)-10.2855, \sigma_{in} = 1.2906, l/L <0.3$<br>$\ln(D)=1.7658Mw-7.8962, \sigma_{in} = 0.9624, l/L \geq 0.3$ | 0.34 |
| | **QUADRATIC**<br>$\ln(D)=1.7895Mw+14.4696(l/L)-20.1723(l/L)^2-10.54512, \sigma_{in} = 1.1346$ | 0.33 |
| | **ELLIPTICAL**<br>$\ln(D)=3.3041\sqrt{1-\frac{1}{0.5^2}[(l/L)-0.5]^2}+1.7927Mw-11.2192, \sigma_{in} = 1.1348$ | 0.33 |
