# Peer review of "Probabilistic Fault Displacement Hazard Analysis for North Tabriz Fault"

_Natural Hazards and Earth System Sciences, 2021_

## Referee Comment (RC2)

[referee-annotated manuscript omitted]

---

## Author Response (AR5)

Subject: Revised manuscript with the reference number: nhess-2021-351

Dear Editor,

We would like to thank you for giving us an opportunity to revise our manuscript. We have addressed all the concerns comments and would like to submit our revised manuscript entitled " **Probabilistic Fault Displacement Hazard Analysis for North Tabriz Fault**" for further consideration. We have modified the manuscript thoroughly by considering all of the comments raised by the reviewers and editor which indeed improved the manuscript. In the following we respond to the comments in details.

Sincerely

Habib

**Comments to the author:**

Dear authors,

I would recommend completing the figure caption of Fig 1 - as the shapes are not explained,
especially the pink-bluish polygons.
Once this is done, it will be accepted.
yours

**OUR ANSWER:**

The caption of Figure 1 was examined, and the details of the figure were explained. In the caption, the location of the fault north of Tabriz and the possible displacement values in the city of Tabriz, and the location of this potential hazard were investigated.

[Figure]

**Figure (1).  The north Tabriz fault and its vicinity to the populated Tabriz city that even a part of this fault passes through the town. The probabilistic displacement area in Tabriz city is shown, assuming a simple fault trace (due to the lack of sufficient instrument data); these possible displacements can be seen up to a distance of ±150 meters from the fault trace in the future.**

**Figure (1) are generated using Google Earth with Digital Globe imagery (© Google Earth 2021).**

We re-examined Figure 1 and used a uniform color to represent Tabriz city and added this to the map legend.
In the end, in your view, we have obtained a suitable color spectrum for Figure 1 for people with poor eyesight through the site (www.color-blindness.com) and item (Blue-Blind/Tritanopia), and we have placed both maps in (answer letter) for you. We hope that we have been able to reach your desired shape.

FIGURES:

[Figure]

*Fig01 with (blindness- Blue-Blind/Tritanopia)*

[Figure]

*Fig01 without (blindness- Blue-Blind/Tritanopia)*

**Previous conversations between us:**

1. Abstract
In this section, we removed inappropriate and extra content, and according to the journal's rules, the number of words used in this section reached less than 200 (words<200).

2. And the contents were placed in the appropriate template for you and sent to you. Also, we sent you all the information, shapes, and code. We hope that we have been able to take a positive step for the hazard analysis community.

1.
Please revise table #1, it seems column5, contains the out-of-place line numbers.

At first, I thought that what you mean is that the numerical values of table (1) are wrong, but finally, I realized that what you mean is a disturbance in the values of the table and the table's subtitle, which collided in the version (along with the change). As a result, we eliminated this disorder, and this weakness has disappeared in the (with and without the Track change versions ). I hope I have reached your goal in this part.

[Figure]

| No. | Cell Size (m²) | a(z) | b(z) | Standard Deviation(s) |
|---|---|---|---|---|
| 1 | 25×25 | -1.1470 | 2.1046 | 1.2508 |
| 2 | 50×50 | -0.9000 | 0.9866 | 1.1470 |
| 3 | 100×100 | -1.0114 | 2.5572 | 1.0917 |
| 4 | 150×150 | -1.0934 | 3.5526 | 1.0188 |
| 5 | 200×200 | -1.1538 | 4.2342 | 1.0177 |

Table 1. distributed different cell sizes (Petersen et al., 2011) Probability of rupture for

2.

For figure 1, please ensure that the colour schemes used in your maps and charts allow readers with colour vision deficiencies to correctly interpret your findings. Please check your figures using the Coblis – Color Blindness Simulator (https://www.color-blindness.com/coblis-color-blindness-simulator/) and revise the colour schemes accordingly.

For this comment of yours, we used the site ((https://www.color-blindness.com/coblis-color-blindness-simulator/) to get figure 1, allowing readers with color vision deficiencies to interpret our findings correctly, and through ( Blue-Blind/Tritanopia and Red-Blind/ Protanopia ) we got two figures for different color spectrum. Finally, we chose the first figure for the article. We have added these two images here so we can discuss them in case of your following comments.

[Figure]

3.
In the tracked changes file there is a non-English text as a comment. As your manuscript language is English please present the comments in English.

We have fixed this weakness for you

4.
I noticed your reference style doesn't meet the NHESS standard, therefore please adjust your reference based on the guideline on this link https://www.natural-hazards-and-earth-system-sciences.net/submission.html#references

We adjusted all the references in the article according to (NHESS standards) and added them in the new version of the article.

1.

e.g. Fig 1a. no georeferencing, quality of figures not really the best, and, a new Discussion chapter that I'd like first to have reviewed.

Answer:

Figure (1a) was removed, and we only used Figure (1b) to show the fault in the north of Tabriz and the city of Tabriz. Figure (1a) was not of good quality. We settled on Figure (1b).

2.

Coloured or marked text in *.pdf manuscript file is not allowed. Please provide a clean version of *pdf manuscript file (with black text) for the next revision.

2. Please ensure that the colour schemes used in your maps and charts allow readers with colour vision deficiencies to correctly interpret your findings. Please check your figures using the Coblis – Color Blindness Simulator (https://www.color-blindness.com/coblis-color-blindness-simulator/) and revise the colour schemes accordingly.

3. Please revise table #1, it seems column5, contains transposed text.

**Answer:** The color of the entire text of the article has been standardized (black), revised, and checked in the table (1). The numerical values were compared with the article by Petersen et al .,

2011 and (table number (4)) in this article, and we did not see any difference in these reference numerical values.

Comments to the author:

Dear authors,
you did not use the right way of tracked changes - what about the 15 references that are not in the text. I do not see the reviewer's annotations, .. how the figures were improved.
This revision has to be completed first.
yours
HB Havenith, assoc. editor.

**Answer:**

3. Abstract:

In this part, one better review of the article's approach was performed, and additional sentences were removed. A critical point added in this section was to refer to following the method and code of the article (Petersen et al., 2011). One of the weaknesses of this article was also mentioned. The primary purpose is to review the article (Peterson et al., 2011) by the literary and ethical rules and reduce the level of error and uncertainty in this field in the next steps of probabilistic displacement risk analysis.

4. Introduction:

The introduction got better structured:
1. The importance of studying earthquakes and hazards due to surface rupture and the importance of experimental relationships according to the study (Stephanie Baiz et al., 2019)
2. The starting point for risk analysis (PFDHA) by (youngs et al. 2003)
3. Study (Petersen et al., 2011) on strike-slip faults and his method
4. Review of several articles by studying the risk analysis (PFDHA) in 2020 and 2021 for the updating of the technical literature of the article

5. Explanations on how to use the method (Petersen et al., 2011) in northwestern Iran and the North Tabriz fault and reviewing the parameters required for this calculation

6. Description of the method of estimating probabilistic outputs for the North Tabriz fault

**Note:** With the addition of several new articles in recent years, a newer study (2020-2021) was performed, and also the input parameters and calculation method were studied.

This section explains the input parameters and that this method is derived from the study (Petersen et al., 2011). We were asked to examine better the parameters used in your previous comment. By adding this paragraph about the importance of studying in the city of Tabriz and the input parameters

5.

 References:

In this section, all additional references were deleted.

6.

In the text of the article and while doing (track changes), we used (annotation) for more detailed explanations, and we hope that the explanations are clear to you.

These margins are highlighted (yellow)

7.

quality of shapes (6-7-8):

In your previous comment, we were told that the quality of the photos (6-7-8) is low, and as a result, we checked it and improved their quality for better clarity. We will send you input data if needed.)

**Previous conversations between us:**

1) The introduction is missing of several worth-to-mention papers and should be improved accordingly.

According to your comment, several papers (mentioned below) have been reviewed and the introduction of the manuscript is revised accordingly. Please see the introduction of the annotated manuscript.

1. Baize, S., Nurminen, F., Sarmiento, A., Dawson, T., Takao, M., Scotti, O., Azuma, T., Boncio, P., Champenois, J., Cinti, F.R. and Civico, R., 2020, A worldwide and unified database of surface ruptures (SURE) for fault displacement hazard analyses: Seismol. Res. Lett, *91(1)*, 499-520.

2. Goda, K., 2021, Potential Fault Displacement Hazard Assessment Using Stochastic Source Models: A Retrospective Evaluation for the 1999 Hector Mine Earthquake: GeoHazards, 2(4), .398-414. https://doi.org/10.3390/geohazards2040022.

3. Nurminen, F., Boncio, P., Visini, F., Pace, B., Valentini, A., Baize, S. and Scotti, O., 2020. Probability of occurrence and displacement regression of distributed surface rupturing for reverse earthquakes: Front. Earth Sci. 8, 456.

4. Katona, T.J., 2020, Safety of Nuclear Power Plants with Respect to the Fault Displacement Hazard. Appl. Sci. 2020, 10, 3624.

2) The input data must be clearly described both in the main text and in the figure. Fault length and selected site are just two examples.

In this section, in addition to re-editing the figures of the manuscript (figure 1), to introduce the input parameters, we also briefly explained these parameters in the introduction. Please see the introduction of the annotated manuscript.

3) The section Methodology of probabilistic fault displacement hazard analysis needs a deep review. I suggest to focus only on the approach used (i.e., Petersen at al.) and I would like to see all equations used ( and how).

Yes – It is done. Please see the " **Methodology of probabilistic fault displacement hazard analysis** " of the annotated manuscript.

4) The section Results and Discussions is very poor. There is no discussion about the results and, for me, it has been very hard to understand how input parameters are considered and which equations are used. This paper could help the seismic hazard local community but it need a major review in order to make the manuscript clear and readable. Reading the paper I have the feeling that most of things are omitted and not well described.

Yes – This section is completely revised according your comments.
To explain the results obtained more clearly, we added Figure 9 which examines and compares the results obtained for two different scenarios and identifies the worst case and will have a better description of the results obtained. Please see the "Results and Discussions" of the annotated manuscript.

[Figure]

(c)

Figure 9. Comparison of drop in displacement values in scenarios, a) 50years, b) 475years, c) 2475years

5) As it is, this work is just an application of an already published work (Petersen et al., 2011). Which is the contribute to the scientific community? even if the authors are in position that they cannot contribute from a methodological point of view, a good discussion section can help to improve the quality of the manuscript, for example, highlighting the critical aspects of this approach, the difficulties that they have found in its application, area that need further and future works, implication in the hazard of the area, and so on.

This comment of yours was very effective and useful for us. we examined the strengths and weaknesses of the work and also examined the applications of this method in northwestern Iran, how they can be effective in the future. In fact, this fault has a high level of risk and lacks high instrumental data and causes uncertainty in studies. For this reason, different scenarios have been considered for displacement estimates.

.

6) Several references (more than 15!!) are in the reference list but they are not in the manuscript. This is a little bit embarrassing. Reference list is important as well figure and main text as.

Yes – we are sorry. All of the mentioned comments are considered and the manuscript is revised accordingly.